# In Vitro Characterization of Periodontal Ligament Stem Cells Derived from Supernumerary Teeth in Three-Dimensional Culture Method

**Yun Yeong Jeong** [1], **Mi Sun Kim** [2], **Ko Eun Lee** [3], **Ok Hyung Nam** [2], **Ji-Hyun Jang** [4], **Sung-Chul Choi** [2] and **Hyo-Seol Lee** [2,*]

1   Department of Pediatric Dentistry, Graduate School, Kyung Hee University, Seoul 02447, Korea; pp5049@naver.com
2   Department of Pediatric Dentistry, School of Dentistry, Kyung Hee University, Seoul 02447, Korea; pedokms@khu.ac.kr (M.S.K.); pedokhyung@khu.ac.kr (O.H.N.); pedochoi@khu.ac.kr (S.-C.C.)
3   Department of Pediatric Dentistry, Kyung Hee University Dental Hospital, Seoul 02447, Korea; olivedlr@naver.com
4   Department of Conservative Dentistry, Kyung Hee University School of Dentistry, Seoul 02447, Korea; jangjihyun@khu.ac.kr
*   Correspondence: snowlee@khu.ac.kr; Tel.: +82-2-958-9363

**Abstract:** Objective: The aim of this study was to compare the characteristics of periodontal ligament stem cells derived from supernumerary teeth (sPDLSCs), cultured using a three-dimensional (3D) method and a conventional two-dimensional (2D) method. Methods: The morphology, viability, and osteogenic differentiation of the cells were analyzed. In addition, gene expression was analyzed by RNA sequencing, to characterize the functional differences. Results: The diameter of the 3D-cultured sPDLSCs decreased over time, but the spheroid shape was maintained for 7 days. The osteogenic differentiation was similar in the 2D and 3D. The gene expression related to the extracellular matrix (7.3%), angiogenesis (5.6%), cell proliferation (4.6%), inflammatory response (3.7%), and cell migration (3.5%) differed ($p < 0.05$). Conclusions: Within the limitations of this study, sPDLSCs varied in formation and function, depending on the culture method. In future, it is necessary to study tissue engineering using the advantages of 3D culture and the fewer ethical problems of supernumerary teeth.

**Keywords:** supernumerary teeth; periodontal ligament stem cell; three-dimensional culture; morphology; osteogenic differentiation; RNA sequencing





## 1. Introduction

Mesenchymal stem cells (MSCs) can be harvested from various human tissues, such as bone marrow, adipose tissue, and umbilical cord blood [1]. In addition, MSCs have been extracted from the periodontal ligaments of permanent teeth, primary teeth, and supernumerary teeth [2–4]. Periodontal ligament (PDL) represents a fibrous network, connecting the cementum of the tooth root and the alveolar bone [5]. It serves many functions, such as tooth support, nutrition, and protection. Periodontal ligament stem cells (PDLSCs) are easy to obtain and have a greater proliferation rate than bone marrow stem cells. PDLSCs are capable of differentiating into osteoblasts, cementoblasts, adipocytes, and chondrocytes, and can form a structure similar to periodontal ligaments in vitro [6–9].

In 2011, PDLSCs were obtained from a supernumerary tooth (sPDLSCs) [4]. The sPDLSCs show a greater colony-forming ability than bone marrow stem cells, and can differentiate into adipocytes and osteoblasts [10]. Expendable dental tissue, such as supernumerary teeth, may be a source of stem cells. Therefore, studies of the biological properties of PDLSCs obtained from various dental tissue are needed to optimize their regenerative effects.

Cell culture is essential for the study of MSCs. In conventional two-dimensional (2D) cell culture methods, cells are cultured on the bottom of a culture-compatible polystyrene plate. Such cultures create an environment for studying biological processes or cell mechanisms under specific experimental conditions, and maintain various types of proliferation [11]. However, in nature, cells grow three-dimensionally (3D) rather than in a 2D plane. Therefore, it is difficult to study cell-to-cell signal transduction and the functions of cells in a conventional 2D culture environment [12]. To overcome these limitations, cell research using 3D culture is under investigation. Three-dimensional cultures have the advantage of emulating the original characteristics of the cell's environment, so that the cell can display form, function, and activity as close to naturally as possible [13,14]. In addition, 3D culture is an alternative to animal experimentation [15].

There are the following two general methods of 3D culture: scaffold and scaffold-free. The scaffold method involves creating a scaffold of microstructures that the cells can use to move, proliferate, differentiate, and remain in contact [15]. However, scaffolds have limitations that can negatively affect cell stability and behavior during incubation [16]. In the scaffold-free method, a pallet or centrifugation tube culture, low-adhesion plate culture, bioreactor, hanging drop culture, or pallet culture and liquid overlay are used to create 3D microstructures without scaffolding [11,15–17]. In these methods, the cells assemble an endogenous extracellular matrix to form their own preferred microenvironment [18,19].

Cells cultured in a 3D environment behave differently than those in a 2D environment [17]. For example, primary articular chondrocytes and hepatocytes cultured in a 2D culture rapidly lose their normal phenotypes, but in 3D cultures the normal phenotype is maintained [20]. In addition, 3D-cultured stem cells exhibit better osteoblastic, adipogenic, and neuronal differentiation [21]. Similarly, 3D-cultured human dental pulp cells show higher expression of osteocalcin, dentin sialophosphoprotein, and alkaline phosphatase [22]. The expression of these proteins is influenced by cell–cell and cell–matrix interactions, which are difficult to imitate in 2D culture. In addition, the expression of genes related to multi-lineage differentiation, such as Nanog and Oct4, as well as those related to pluripotency transcription factors, is higher in 3D-cultured cells. These studies demonstrate that cell culture conditions influence gene expression [17,23]. However, previous studies have usually used microarrays and polymerase chain reaction, and few studies have analyzed genome-wide expression.

The aim of this study was to compare the characteristics of sPDLSCs cultured using a 3D and 2D method. Morphology, viability, osteogenic differentiation, and gene expression were evaluated. These experiments would reveal what physical and functional properties the same cells would have in different culture systems.

## 2. Materials and Methods

### 2.1. 2D and 3D Culture of Human sPDLSCs

sPDLSCs were purchased from Cell Engineering for Origin (Seoul, Korea). They were cultured in a 37 °C incubator with an atmosphere containing 5% $CO_2$ in Dulbecco's modified Eagle's medium (DMEM; Gibco, Waltham, MA, USA). After three to four passages, the cells were split and thereafter grown in 2D or 3D cultures.

The formation of stem cell spheroids in 3D culture was performed with Stemfit 3D (Prosys Stemfit 3D®, Prodizen, Seoul, Korea; Figure 1). Cells ($1.2 \times 10^6$) were seeded in a concave micromold and cultured in DMEM in a 37 °C incubator with an atmosphere containing 5% $CO_2$.

### 2.2. Morphology

The 2D-cultured cells and the spheroids formed in 3D culture were observed at 1, 3, 5, and 7 days under a fluorescence microscope (JuLI; Nanoentek, Seoul, Korea).

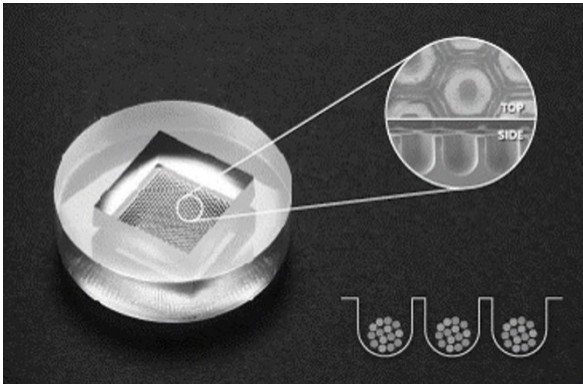

**Figure 1.** Stemfit 3D® uses a non-adhesive plate with a polydimethylsiloxane-based concave micro-mold 600 μm in diameter. It can control the size of spheroids and enhance their long-term stability.

*2.3. Assays of Cell Viability*

The cell counting kit-8 assay (CCK-8; Dojindo Molecular Technologies, Kumamoto, Japan) and live/dead viability/cytotoxicity kit for mammalian cells (Invitrogen, Carlsbad, CA, USA) were used to quantify cell proliferative ability. The CCK-8 assay is based on the activation of 1-methoxy phenazinium methylsulfate by nicotinamide adenine dinucleotide phosphate (NADP) and nicotinamide adenine dinucleotide phosphate hydrogen (NADPH) produced by dehydrogenase in living cells. Water-soluble tetrazolium salt, 2-(2-methoxy-4-nitrophenyl)-3-(4-nitrophenyl)-5-(2,4-disulfophenyl)-2H tetrazolium, and monosodium salt (WST-8) are reduced to orange formazan; therefore, the amount of WST-8 formazan produced by dehydrogenases is directly proportional to the number of viable cells.

The live/dead assay quantifies living and dead cells by measuring their esterase activity and plasma membrane integrity. The non-fluorescent dye calcein AM is converted into a fluorescent calcein that fluoresces in the presence of esterase in living cells, producing a uniform green fluorescence. Ethidium homodimer-1 enters dead cells through damaged membranes and binds with nucleic acids to produce red fluorescence.

The 2D- and 3D-cultured sPDLSCs were seeded at a density of $1.0 \times 10^5$ per well on a six-well plate and subjected to live/dead assay on days 3, 5, and 7. Images were obtained with a fluorescence microscope (IX71; Olympus, Tokyo, Japan). CCK-8 assay was performed on days 1, 3, 5, and 10, and the absorbance at 450 nm was measured with a Benchmark Plus multiplate spectrophotometer (Bio-Rad, Hercules, CA, USA).

*2.4. Osteogenic Differentiation*

After 2D or 3D culture, sPDLSCs were cultured in osteo-inductive medium (osteo group) or basal medium (control group) to confirm their ability to differentiate into bone cells. The osteo group was cultured for 2 weeks in osteo-inductive medium with 15% fetal bovine serum, 250 μL gentamycin reagent solution (final concentration 5 μg/mL), 5 mL filtered *L*-ascorbic acid (final concentration 10 mM), and 5 mL filtered dexamethasone. Alizarin red S (ARS) staining was performed to visualize calcium formation for evaluating differentiation. Cells were fixed in ice-cold 70% ethanol for 15 min, stained with ARS solution for 3 min, washed, and observed under an optical microscope. For quantitative evaluation of hard tissue formation, 10% cetylpyridinium chloride was added for 10 min, and the ARS stain was extracted and transferred to a 96-well plate. The absorbance was measured at a wavelength of 562 nm with a Benchmark Plus multiplate spectrophotometer (Bio-Rad).

*2.5. Gene Expression*

2.5.1. RNA Isolation

RNA was isolated from 2D- and 3D-cultured sPDLSCs on day 5 with RiboEx (GeneAll, Seoul, Korea), and RNA quality was assessed with an Agilent 2100 bioanalyzer with the RNA 6000 nano chip (Agilent Technologies, Amstelveen, The Netherlands). RNA

quantification was performed with an ND-2000 spectrophotometer (Thermo, Waltham, DE, USA).

### 2.5.2. Library Preparation and Sequencing

Libraries of control and test RNA were constructed with the QuantSeq 3′ mRNA-Seq library prep kit (Lexogen, Vienna, Austria) according to the manufacturer's instructions. In brief, 500 ng RNA was prepared, hybridized to an oligo-dT primer containing an Illumina-compatible sequence at its 5′ end, and subjected to reverse transcription. After degradation of the RNA template, second-strand synthesis was initiated with a random primer containing an Illumina-compatible linker sequence at its 5′ end. The double-stranded library was purified with magnetic beads to remove all reaction components. Next the library was amplified to add the complete adapter sequences required for cluster generation. The finished library was purified to remove PCR components. High-throughput sequencing was performed as single-end 75 bp sequencing using a NextSeq 500 sequencer (Illumina, San Diego, CA, USA).

### 2.5.3. Data Processing for the Identification of Differentially Expressed Genes

QuantSeq 3′ mRNA-Seq reads were aligned with Bowtie 2. Bowtie 2 indices were generated from a genome assembly sequence or the representative transcript sequences for aligning the genome and transcriptome. The alignment file was used to assemble transcripts, estimate their abundance, and detect the differential expression of genes. Differentially expressed genes were identified based on counts from unique and multiple alignments using coverage in BEDtools. The read count data were processed based on the quantile normalization method with EdgeR in R (R Development Core Team, Vienna, Austria) with Bioconductor.

### 2.6. Statistical Analyses

Statistical analyses were performed with SPSS version 25.0 (IBM, Armonk, NY, USA). One-way analysis of variance (ANOVA) and the Scheffé post hoc test were performed to compare absorbance values. $p < 0.05$ was considered indicative of statistical significance.

### 3. Results

### 3.1. Morphology

The sPDLSCs grown in the 2D culture exhibited a bipolar and stellate form, were attached to the plate, and increased in number over time (Figure 2A–D). The sPDLSCs grown in the 3D culture aggregated and became spheroid for about 24 h (Figure 2E). Although the diameters of the spheroids decreased significantly over the first 5 days (75.3%), they maintained their shape (Figure 2E–H).

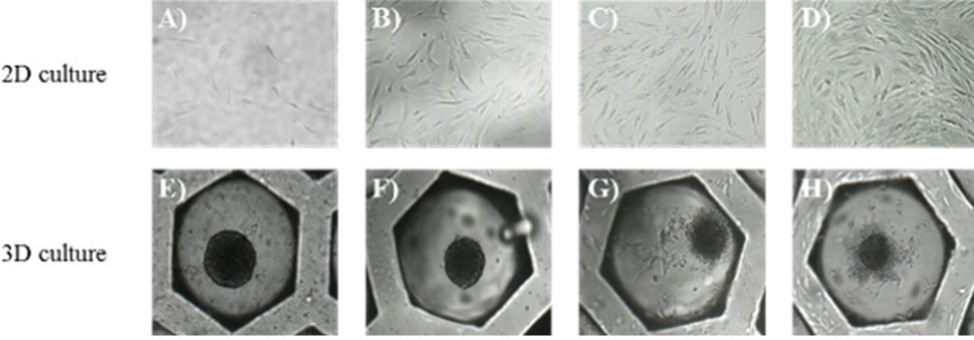

**Figure 2.** Morphology, days 1–7. (**A–D**) Morphology of sPDLSCs at days 1, 3, 5, and 7 in 2D cultures (magnification, ×200). The sPDLSCs increased in number over time. (**E–H**) Spheroid of sPDLSCs at days 1,3,5, and 7 in 3D cultures (magnification, ×200). The sPDLSCs aggregated and became spheroid for about 24h (**E**). Diameters of the spheroids decreased significantly over the time.

### 3.2. Cell Viability

The absorbance of 2D-cultured sPDLSCs increased significantly over time, from 0.26 to 2.60 (Figure 3). In particular, it increased exponentially on day 10 ($p < 0.05$). By contrast, the absorbance of 3D-cultured sPDLSCs decreased over time, from 0.31 to 0.25. This decrease was significant from days 3 to 5 ($p < 0.05$), but not between days for the first 3 days ($p > 0.05$). At day 10, the absorbance increased slightly, but not significantly ($p > 0.05$).

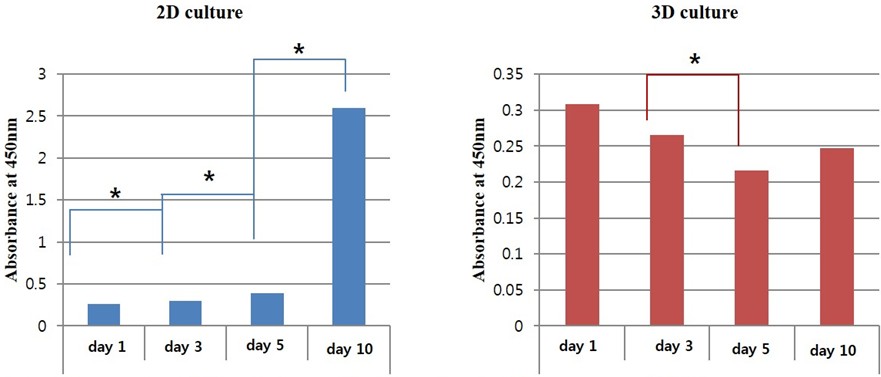

\* $p < 0.05$ comparison with the control group; Schaffe's *post hoc* test following one-way ANOVA.

**Figure 3.** Results of CCK-8 assays. The 2D-cultured sPDLSCs showed an increase in absorbance and 3D-cultured sPDLSCs showed a significant decrease in absorbance from day 1 to day 5. The absorbance of 3D-cultured sPDLSCs increased non-significantly on day 10. \* $p < 0.05$; Scheffé post hoc test following one-way ANOVA.

The 2D-cultured sPDLSCs did not show a significant decrease in viability; however, the cell viability decreased significantly in the center of the spheroids in the 3D culture (Figure 4).

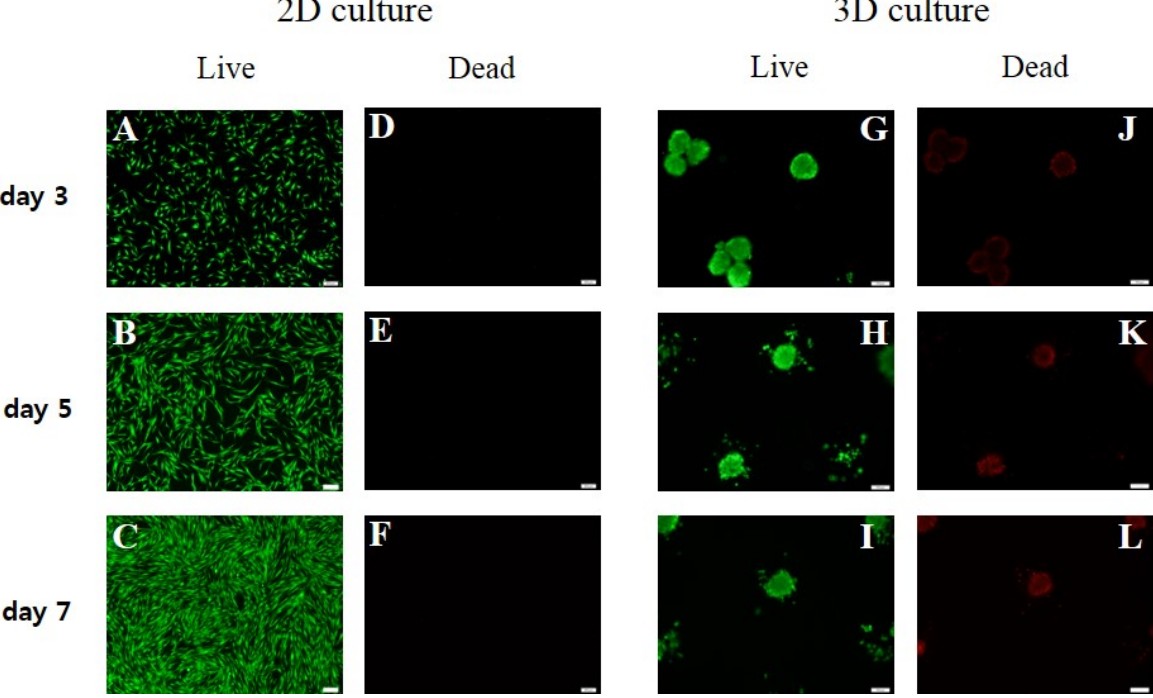

**Figure 4.** Live/dead assays. The 2D-cultured sPDLSCs did not show a significant decrease in viability; however, the cell viability decreased significantly in the center of the spheroids in the 3D culture. (**A–F**) Live/dead assay results for 2D-cultured sPDLSCs at days 3, 5, and 7 (magnification, ×100). The number of cells increased, and no dead cells were observed. (**G–L**) Live/dead assay results for 3D-cultured sPDLSCs at days 3, 5, and 7 (magnification, ×100). The number of dead cells in the center of spheroids increased with decreasing spheroid diameter.

### 3.3. Stemness and Osteogenic Differentiation

ARS staining revealed significant extracellular calcium deposits in both the 2D and 3D cultures, which was indicative of osteogenic differentiation (Figure 5). Compared to the control group, the absorbance of the ARS stain extract was 2.3 in the 3D group, which was slightly higher than in the 2D group.

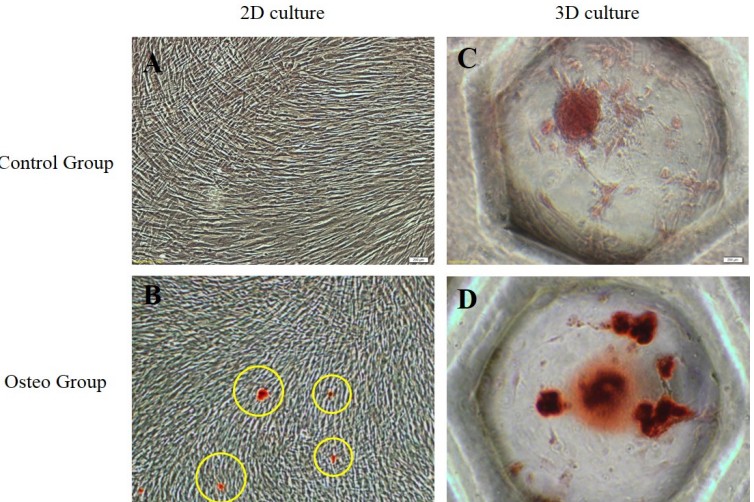

**Figure 5.** Evaluation of osteogenesis in 2D- and 3D-cultured sPDLSCs (magnification, ×200). (**A**) Control group of 2D-cultured sPDLSCs in basal media. (**B**) Osteo group of 2D-cultured sPDLSCs in osteogenic media. ARS staining revealed significant extracellular calcium deposits (yellow circles). (**C**) Control group of 3D-cultured sPDLSCs in basal media. (**D**) Osteo group of 3D-cultured sPDLSCs in osteogenic media.

### 3.4. Identification of Differentially Expressed Genes and Gene Ontology Analyses

The expression of 25,737 differentially expressed genes in the 2D- and 3D-cultured sPDLSCs was examined. Of these, 5664 genes, related to cell function, immunity, inflammation, and the extracellular matrix, were selected for further analyses. The expression of 89 genes (58.9%) was upregulated, and that of 62 genes (41.1%) was downregulated in the 3D culture compared to in the 2D culture. The genes up- or downregulated were related to the extracellular matrix (ECM; 7.3%), angiogenesis (5.6%), cell proliferation (4.6%), inflammatory response (3.7%), and cell migration (3.5%; Figure 6). The genes in the 3D-cultured sPDLSCs that showed at least a 50-fold increase or decrease in expression compared to the 2D-cultured sPDLSCs are listed in Tables 1 and 2.

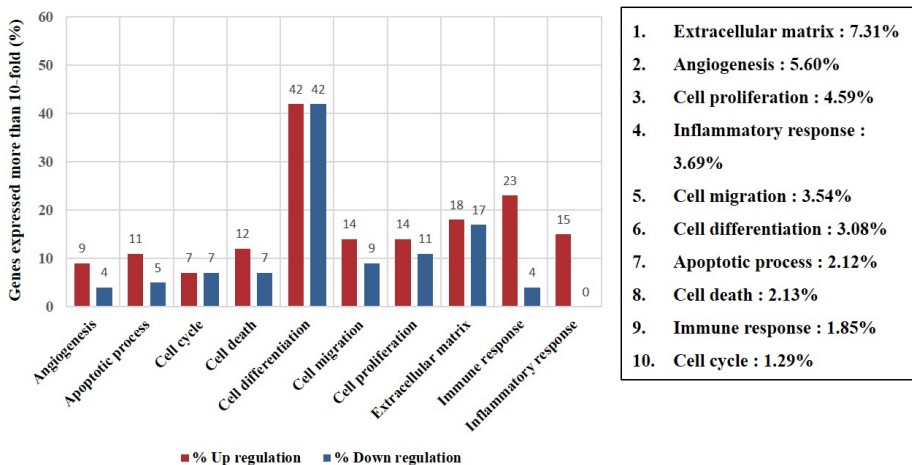

**Figure 6.** Expression of filtered gene categories. Distribution and ratio of genes in 10 selected categories with 10-fold increased expression ($p < 0.05$).

**Table 1.** Genes upregulated more than 10-fold in 3D- compared to 2D-cultured sPDLSCs.

| Gene Name | Fold Change | Description | Related Function |
|---|---|---|---|
| AREG | 1172.7 | Amphiregulin | Cell differentiation, cell proliferation |
| SMOC1 | 1161.3 | SPARC-related modular calcium binding 1 | Extracellular matrix |
| MMP1 | 829.2 | Matrix metallopeptidase 1 | Cell migration, extracellular matrix |
| STC1 | 791.3 | Stanniocalcin 1 | Cell differentiation, cell proliferation |
| MMP10 | 349.8 | Matrix metallopeptidase 10 | Extracellular matrix |
| PGF | 341.9 | Placental growth factor | Angiogenesis |
| PTGS2 | 195.6 | Prostaglandin-endoperoxide synthase 2 | Angiogenesis, cell differentiation, inflammatory response |
| COL5A3 | 173.4 | Collagen type V alpha 3 | Extracellular matrix |
| IL24 | 156.0 | Interleukin 24 | Apoptotic process, cell death |
| FRZB | 119.4 | Frizzled-related protein | Cell differentiation |
| BMP2 | 112.8 | Bone morphogenetic protein 2 | Cell differentiation, cell proliferation, inflammatory response |
| SFRP2 | 104.5 | Secreted frizzled-related protein 2 | Angiogenesis, apoptotic process, cell death, cell differentiation, cell proliferation, extracellular matrix |
| FGL2 | 98.6 | Fibrinogen-like 2 | Immune response |
| IER3 | 97.3 | Immediate early response 3 | Apoptotic process, cell death |
| TFPI2 | 94.9 | Tissue factor pathway inhibitor 2 | Extracellular matrix |
| DPT | 83.6 | Dermatopontin | Extracellular matrix |
| NR4A2 | 82.5 | Nuclear receptor subfamily 4 group A member 2 | Cell differentiation, cell migration |
| TNC | 82.1 | Tenascin C | Cell differentiation, extracellular matrix |
| NPNT | 73.8 | Nephronectin | Cell differentiation, extracellular matrix |
| NDP | 64.7 | Norrie disease (pseudoglioma) | Cell proliferation, extracellular matrix |
| ACKR4 | 64.5 | Atypical chemokine receptor 4 | Immune response |
| IL18R1 | 63.5 | Interleukin 18 receptor 1 | Cell differentiation, immune response |
| MMP8 | 62.5 | Matrix metallopeptidase 8 | Extracellular matrix |
| SPRY1 | 61.6 | Sprouty RTK signaling antagonist 1 | Cell cycle, cell differentiation |
| RGS2 | 61.4 | Regulator of G-protein signaling 2 | Cell cycle, cell differentiation |
| IL1R1 | 60.3 | Interleukin 1 receptor type 1 | Immune response |
| PTGS1 | 60.3 | Prostaglandin-endoperoxide synthase 1 | Inflammatory response |
| SPON1 | 53.6 | Spondin 1 | Extracellular matrix |
| EDNRB | 52.4 | Endothelin receptor type B | Cell differentiation, cell migration |
| PECAM1 | 49.3 | Platelet and endothelial cell adhesion molecule 1 | Angiogenesis, cell differentiation, cell migration |
| IL1RN | 46.1 | Interleukin 1 receptor antagonist | Immune response, inflammatory response |
| PRDM1 | 44.6 | PR domain 1 | Cell differentiation, cell proliferation, immune response |
| CHST2 | 42.1 | Carbohydrate sulfotransferase 2 | Inflammatory response |
| APOE | 40.0 | Apolipoprotein E | Cell differentiation, extracellular matrix |
| NOX4 | 38.8 | NADPH oxidase 4 | Cell differentiation, inflammatory response |
| SNAP25 | 38.2 | Synaptosome-associated protein 25 kDa | Cell differentiation, immune response |
| TNN | 36.8 | Tenascin N | Cell differentiation, extracellular matrix |
| VAV3 | 34.8 | VAV guanine nucleotide exchange factor 3 | Angiogenesis, cell migration |
| BCL2A1 | 31.5 | BCL2-related protein A1 | Apoptotic process, cell death |
| JUP | 30.3 | Junction plakoglobin | Cell death, cell differentiation, cell migration, extracellular matrix, immune response |
| AKR1C1 | 30.3 | Aldo-keto reductase family 1, member C1 | Cell differentiation |
| NRN1 | 29.8 | Neuritin 1 | Cell differentiation |
| ITGA2 | 28.6 | Integrin subunit alpha 2 | Cell differentiation, cell migration, cell proliferation |
| SLC1A3 | 28.3 | Solute carrier family 1 member 3 | Cell differentiation |

**Table 1.** *Cont.*

| Gene Name | Fold Change | Description | Related Function |
|---|---|---|---|
| SECTM1 | 27.7 | Secreted and transmembrane 1 | Immune response |
| CD7 | 27.2 | CD7 molecule | Immune response |
| AKR1C2 | 27.1 | Aldo-keto reductase family 1, member C2 | Cell differentiation |
| AIM2 | 26.7 | Absent in melanoma 2 | Apoptotic process, cell death, immune response, inflammatory response |
| GPM6B | 26.5 | Glycoprotein M6B | Cell differentiation |
| ETV1 | 26.0 | ETS Variant 1 | Cell differentiation |
| CXCL8 | 24.7 | C-X-C motif chemokine ligand 8 | Angiogenesis, cell cycle, cell migration, immune response, inflammatory response |
| NKX2-5 | 24.5 | NK2 homeobox 5 | Apoptotic process, cell death, cell differentiation, cell proliferation |
| TNFAIP6 | 24.5 | TNF-alpha-induced protein 6 | Immune response, inflammatory response |
| NEFM | 23.6 | Neurofilament, medium polypeptide | Cell differentiation |
| CCL2 | 23.3 | C-C motif chemokine ligand 2 | Cell migration, immune response |
| KYNU | 22.7 | Kynureninase | Immune response |
| RANBP3L | 22.5 | RAN-binding protein 3-like | Cell cycle, cell differentiation |
| NTN1 | 21.6 | Netrin 1 | Apoptotic process, cell death, cell differentiation, cell migration, extracellular matrix |
| SEPT3 | 21.0 | Septin 3 | Cell cycle |
| GABRB2 | 20.4 | Gamma-aminobutyric acid type A receptor beta2 subunit | Cell differentiation |
| EDNRA | 20.3 | Endothelin receptor type A | Angiogenesis, cell differentiation, cell proliferation |
| CHST1 | 19.5 | Carbohydrate sulfotransferase 1 | Inflammatory response |
| AFAP1L2 | 19.5 | Actin filament-associated protein 1-like 2 | Inflammatory response |
| CRYGD | 19.4 | Crystallin, gamma D | Cell differentiation |
| CMKLR1 | 18.6 | Chemerin chemokine-like receptor 1 | Immune response |
| WNT7B | 17.4 | Wnt family member 7B | Cell differentiation, cell proliferation, extracellular matrix |
| PTPRU | 16.5 | Protein tyrosine phosphatase, receptor type U | Cell differentiation |
| ITGB3 | 16.4 | Integrin subunit beta 3 | Angiogenesis, cell differentiation, cell migration |
| TNFRSF21 | 15.5 | Tumor necrosis factor receptor superfamily member 21 | Apoptotic process, cell death, cell differentiation, immune response |
| NYAP1 | 14.4 | Neuronal tyrosine phosphorylated phosphoinositide-3-kinase adaptor 1 | Cell differentiation |
| GRIN2A | 13.3 | Glutamate ionotropic receptor NMDA type subunit 2A | Cell differentiation |
| PHOX2B | 12.8 | Paired-like homeobox 2b | Cell differentiation, cell migration |
| GPR68 | 12.8 | G-protein-coupled receptor 68 | Inflammatory response |
| FPR1 | 12.7 | Formyl peptide receptor 1 | Cell migration, immune response, inflammatory response |
| EREG | 12.6 | Epiregulin | Angiogenesis, cell cycle, cell differentiation, cell proliferation |
| FGF7 | 12.5 | Fibroblast growth factor 7 | Cell proliferation |
| TNFSF10 | 12.1 | Tumor necrosis factor superfamily member 10 | Apoptotic process, cell death, immune response |
| LRP4 | 11.5 | LDL receptor-related protein 4 | Cell differentiation |
| PTPN22 | 11.3 | Protein tyrosine phosphatase, non-receptor type 22 | Cell differentiation |
| RNF152 | 11.3 | Ring finger protein 152 | Apoptotic process, cell death |

**Table 1.** *Cont.*

| Gene Name | Fold Change | Description | Related Function |
|---|---|---|---|
| SOCS2 | 11.3 | Suppressor of cytokine signaling 2 | Cell differentiation |
| CXCL3 | 11.2 | CXC motif chemokine ligand 3 | Cell migration, immune response, inflammatory response |
| GEM | 11.1 | GTP-binding protein overexpressed in skeletal muscle | Cell cycle, immune response |
| ZNF443 | 10.8 | Zinc finger protein 443 | Apoptotic process, cell death |
| MGP | 10.7 | Matrix Gla protein | Extracellular matrix |
| TNIK | 10.5 | TRAF2 and NCK interacting kinase | Cell differentiation |
| NR4A3 | 10.3 | Nuclear receptor subfamily 4 group A member 3 | Cell differentiation, cell proliferation, immune response |
| CSGALNACT1 | 10.3 | Chondroitin sulfate N-acetylgalactosaminyltransferase 1 | Cell proliferation |
| CTSS | 10.3 | Cathepsin S | Immune response |

**Table 2.** Genes downregulated >10-fold in 3D- compared to 2D-cultured sPDLSCs.

| Gene Name | Fold Change | Description | Related Function |
|---|---|---|---|
| ACAN | 177.4 | Aggrecan | Cell differentiation, extracellular matrix |
| NGF | 136.7 | Nerve growth factor | Apoptotic process, cell death, cell differentiation |
| HAPLN1 | 106.0 | Hyaluronan and proteoglycan link protein 1 | Extracellular matrix |
| NPR3 | 102.8 | Natriuretic peptide receptor 3 | Cell proliferation |
| CTNND2 | 86.6 | Catenin delta 2 | Cell differentiation |
| ADM2 | 81.4 | Adrenomedullin 2 | Angiogenesis |
| TRIM67 | 58.4 | Tripartite motif containing 67 | Cell differentiation |
| ADAMTS14 | 55.6 | ADAM metallopeptidase with thrombospondin type 1 motif 14 | Extracellular matrix |
| DLGAP5 | 52.7 | Discs large homolog-associated protein 5 | Cell cycle, cell proliferation |
| ADRA1B | 45.1 | Adrenoceptor alpha 1B | Cell proliferation |
| LGR5 | 45.1 | Leucine-rich repeat-containing G-protein-coupled receptor 5 | Cell differentiation, cell proliferation |
| TENM2 | 41.3 | Teneurin transmembrane protein 2 | Cell differentiation |
| SLITRK1 | 41.2 | SLIT and NTRK like family member 1 | Cell differentiation |
| MAP2 | 38.3 | Microtubule associated protein 2 | Cell differentiation |
| IL7R | 35.5 | Interleukin 7 receptor | Cell differentiation, cell proliferation, immune response |
| GLI1 | 33.3 | GLI family zinc finger 1 | Cell differentiation |
| KRT33B | 32.6 | Keratin 33B | Cell death, cell differentiation |
| COL1A1 | 27.8 | Collagen type I alpha 1 | Cell differentiation, cell migration, extracellular matrix |
| HILS1 | 25.5 | Histone linker H1 domain, spermatid-specific 1 (pseudogene) | Cell differentiation |
| ANK2 | 25.5 | Ankyrin 2, neuronal | Cell differentiation |
| PRELP | 25.0 | Proline/arginine-rich end leucine-rich repeat protein | Cell differentiation, extracellular matrix |
| KRT34 | 23.2 | Keratin 34 | Cell death, cell differentiation |
| TTK | 19.6 | TTK protein kinase | Cell cycle |
| ASB2 | 18.7 | Ankyrin repeat and SOCS box-containing 2 | Cell differentiation |
| PSG1 | 17.8 | Pregnancy-specific beta-1-glycoprotein 1 | Cell migration |
| ASPN | 16.9 | Asporin | Extracellular matrix |
| RSPO2 | 16.5 | R-Spondin 2 | Cell differentiation |

**Table 2.** *Cont.*

| Gene Name | Fold Change | Description | Related Function |
|---|---|---|---|
| TAGLN | 15.7 | Transgelin | Cell differentiation |
| FBN2 | 15.3 | Fibrillin 2 | Extracellular matrix |
| LOX | 15.1 | Lysyl oxidase | Extracellular matrix |
| SEMA3D | 14.4 | Semaphorin 3D | Cell differentiation |
| LEP | 14.0 | Leptin | Cell differentiation, cell migration |
| COL6A6 | 13.6 | Collagen type VI alpha 6 | Extracellular matrix |
| MMP15 | 13.6 | Matrix metallopeptidase 15 | Extracellular matrix |
| CREB3L1 | 13.5 | cAMP-responsive element-binding protein 3-like 1 | Cell differentiation |
| EFHD1 | 13.5 | EF-hand domain family member D1 | Cell differentiation |
| KRTAP1-5 | 13.4 | Keratin-associated protein 1-5 | Cell differentiation |
| ANLN | 13.3 | Anillin actin-binding protein | Cell cycle, cell differentiation, cell migration |
| HLA-DMA | 13.1 | Major histocompatibility complex, class II, DM alpha | Immune response |
| MKI67 | 13.0 | Marker of proliferation Ki-67 | Cell cycle, cell proliferation |
| GDF6 | 12.9 | Growth differentiation factor 6 | Cell death, cell differentiation |
| ANKRD1 | 12.7 | Ankyrin repeat domain 1 | Cell differentiation |
| BCL2 | 12.6 | B-cell CLL/lymphoma 2 | Cell death, cell differentiation, cell proliferation |
| RIMS1 | 12.5 | Regulating synaptic membrane exocytosis 1 | Cell differentiation |
| ECM2 | 12.4 | Extracellular matrix protein 2 | Cell differentiation, extracellular matrix |
| TUBB2B | 12.2 | Tubulin beta 2B class IIb | Cell differentiation, cell migration |
| CLDN1 | 12.0 | Claudin 1 | Cell differentiation, immune response |
| AURKB | 11.7 | Aurora kinase B | Cell cycle |
| RTN1 | 11.6 | Reticulon 1 | Cell differentiation |
| BCAT1 | 11.6 | Branched chain amino acid transaminase 1 | Cell cycle, cell proliferation |
| KIRREL3 | 11.5 | Kin of IRRE-like 3 (*Drosophila*) | Cell differentiation, cell migration |
| SLC4A5 | 11.2 | Solute carrier family 4 member 5 | Cell differentiation |
| MYC | 11.1 | v-Myc avian myelocytomatosis viral oncogene homolog | Cell cycle, cell differentiation, cell proliferation, immune response |
| CYR61 | 11.0 | Cysteine-rich angiogenic inducer 61 | Angiogenesis, apoptotic process, cell death, cell differentiation, cell migration, cell proliferation, extracellular matrix |
| CCBE1 | 11.0 | Collagen and calcium-binding EGF domains 1 | Angiogenesis, extracellular matrix |
| TPM1 | 10.6 | Tropomyosin 1 (alpha) | Cell differentiation |
| CHAC1 | 10.5 | ChaC glutathione-specific gamma-glutamylcyclotransferase 1 | Apoptotic process, cell death, cell differentiation |
| CTGF | 10.4 | Connective tissue growth factor | Angiogenesis, cell differentiation, cell migration, cell proliferation, extracellular matrix |
| NAV2 | 10.3 | Neuron navigator 2 | Extracellular matrix |
| ETV7 | 10.2 | ETS Variant 7 | Cell differentiation |
| CLEC3B | 10.1 | C-type lectin domain family 3 member B | Extracellular matrix |
| MATN2 | 10.1 | Matrilin 2 | Cell differentiation, cell migration, extracellular matrix |

## 4. Discussion

A scaffold-free 3D culture method, Stemfit 3D®, was used for the 3D culture of sPDLSCs, and was compared to a traditional 2D culture. The analyses of morphology, viability, osteogenic differentiation, and gene expression confirmed that the functions and characteristics of the sPDLSCs varied depending on the culture method.

Stemfit 3D® uses a non-adhesive plate, which is 600 μm in diameter, with a poly dimethylsiloxane-based concave micromold. Concave micromolds were used because they

accelerate cell aggregation compared to other methods (plane, cylindrical), and the cells form spheroids of uniform size that are easily harvested [2]. Moreover, 3D culture methods that use low-adhesive plates have lower long-term stability than other methods, because of the difficulty limiting the size of the spheroids [21]. However, because Stemfit 3D® can control the size of the spheroids, it shows improved stability over a longer period. Because of its economy, its ease of use, and the stability of the spheroids, Stemfit 3D® is commonly used in stem cell cytology and studies of other types of cells [3,4].

The diameter of the sPDLSC spheroids in the 3D culture decreased (75.3%) over the first 5 days, and decreased more slowly thereafter. The initial reduction in diameter was caused by cell aggregation and organization, but the subsequent reduction appears to have been a result of restricted nutrients and oxygen. Several models for the transport of nutrients, oxygen, and waste in spheroids have been verified [8]. When the diffusion of oxygen is limited, avascular tissue forms, and inefficient transport results in the accumulation of metabolic waste and the formation of a necrotic core at the center of the spheroid [10,12,14]. Several studies on MSC spheroids, including ones by Hildebrandt et al. [14], who investigated human MSCs, and by Yamaguchi et al. [18], who evaluated rat MSCs, have reported that spheroids form within 1 day and that their diameters decrease over time. However, Lee et al. [19] reported that the diameter of human dental pulp stem cell spheroids, cultured on a six-well non-adhesive plate, increased over time. This discrepancy is due to the different culture methods used. Hildebrandt et al. [14] and Yamaguchi et al. [18] independently cultured MSCs in a small volume of medium, or used the scaffold method, whereas Lee et al. [19] used a relatively large volume of medium. It is possible that spheroid–spheroid interactions may increase the diameter of the spheroids, but further study is needed.

CCK-8 and live/dead assays were used to confirm the proliferative ability and viability of 3D-cultured sPDLSCs; these are frequently used in 2D culture experiments [22,23]. The CCK-8 assay revealed that the absorbance of the 2D culture increased over time, and that of 3D culture tended to decrease over time. This was probably due to continued proliferation of cells in the 2D environment, whereas in the 3D environment the differentiation rate decreased as a result of restricted nutrients and oxygen in the center of the spheroids [20]. In addition, in the 3D culture, the number of dead cells in the center of the spheroids increased slightly over time. Similarly, in one study, spheroids of human MSCs in pallet culture maintained a stable internal structure for one month. However, a necrotic area developed in the center of the spheroids, with a loss of proliferation, impaired structural stability, and decreased cell-to-cell contact at two months [20]. Although quantitatively evaluating 2D and 3D culture methods is difficult [24], these results support the notion that the characteristics of 3D-cultured cells are fundamentally different from those of 2D-cultured cells.

Three-dimensional culture improves osteogenic differentiation compared to 2D culture. In addition, the intercellular interactions of embryonic stem cells are enhanced in a 3D environment [25]. For example, progenitor cells derived from the salivary gland can differentiate into hepatocytic and pancreatic islet cell lineages only when cultured in a 3D environment [26]. For this reason, 3D culture has attracted attention in stem cell biology and oncology, as well as in dentistry [12]. For example, 3D-cultured dental pulp stem cells or gingiva/papilla-derived stem cells show higher ALP activity than those grown in 2D culture [3,6,7]. In addition, the expression of genes associated with bone formation, such as BMP2, RUNX2, and dentine sialophosphoprotein, is increased in 3D-cultured cells [19].

The expression of BMP2 was upregulated 112.8-fold in 3D culture compared to in 2D culture. However, compared to our previous studies, the expression of osteogenic differentiation factors, such as RUNX2 and BMP2, tended to be lower in sPDLSCs than in pPDLSCs. Therefore, sPDLSCs can differentiate in 2D and 3D cultures, but their osteogenic differentiation is lower than that of cells extracted from permanent teeth.

RNA sequencing was conducted to analyze the gene expression profiles of sPDLSCs grown in 2D and 3D cultures. Analyses of gene expression profiles by microarray or PCR

are limited by the fact that only selected genes can be identified. Because RNA sequencing captures a wider range of active genes, it is possible to detect small changes in expression with a limited amount of transcript [27].

The RNA sequencing results showed that the expression of genes related to the ECM, angiogenesis, cell proliferation, and the inflammatory response (such as AREG, SMOC1, MMP1, STC1, MMP10, PGF, and PTGS2) was very high. Indeed, 3D-cultured MSCs reportedly exhibit increased expression of genes related to angiogenesis, inflammation, and proliferation, such as TGS-6, IL24, VEGF, FGF-2, CXCR4, MCP3, RANTES, EGF, and SDF [21]. The increased expression of angiogenic genes is confirmed by the observation that 3D-cultured MSCs had greater angiogenic effects than 2D-cultured MSCs, enhancing tissue regeneration [28]. In addition, 3D-cultured MSCs show increased expression of genes related to osteogenesis and adipogenesis, such as RUNX-2, BMP2, OXC, OC, OPN, COLI, OCN, DDPP, DMP1, LPL, FABP4, PLN4, and PPARr [15,18,19]. These genes promote adipogenic and osteogenic differentiation by increasing the production of ECM proteins and endogenous growth factors.

The initial coalescence of spheroids is caused by caspase-dependent IL-1 autocrine signaling, which upregulates EGR2 [16]. In this study, the 3D-cultured sPDLSCs showed upregulated expression of EGR2 (2.3-fold). In addition, expression of EGR1 and EGR3 increased 5.4- and 9.7-fold, respectively. Similarly, immunomodulators, such as TN-FAIP6 (24.5-fold), are reportedly upregulated in response to IL1 autocrine signaling in 3D spheroids [29,30]. In the center of spheroids, where the apoptosis zone is formed as a result of nutrient and oxygen limitation, hypoxia- and apoptosis-related genes, such as VEGFA and hypoxia-inducible factor (HIF), are upregulated. In this study, hypoxia-associated genes, such as VEGFA (6.1-fold) and HIF3A (2.0-fold), and apoptosis-related genes, such as IL24 (156-fold) and COMP (SFRP2), were highly upregulated in the spheroids. In addition, the expression of genes related to angiogenesis, such as PGF (341-fold) and PTGS2 (195.6-fold), was highly upregulated. These results are consistent with the observation that over time, the apoptotic zone of the spheroid increased and the surface proliferation zone decreased.

This study has several limitations. First, the decreased diameter and reduced proliferative ability of the 3D-cultured sPDLSCs may have been a result of their inherent characteristics or the morphologic characteristics of the spheroid, which hamper the supply of nutrients and oxygen, and the discharge of waste products. Our results are compatible with those of other studies that used 3D pallet cultures, low-adhesive plates, or scaffolds. In this regard, it may be necessary to compare other 3D methods, such as a bioreactor. Second, it may not have been optimal to use assays optimized for a 2D environment to evaluate 3D-cultured cells. For example, the CCK-8 assay involves measuring the color change caused by the dehydrogenation of water-soluble tetrazolium salts by living cells. Therefore, it is not ideal for comparing the absorbance of 2D and 3D cultures quantitatively, because the number of cells contacting the solution differs. It is difficult, or impossible, to find protocols and assays optimized for 3D-cultured sPDLSCs, so further research is needed to establish optimized methods for analyzing 3D-cultured cells. Third, this was a pilot study of 2D- and 3D-cultured sPDLSCs, and so the sample was small. Fourth, several prior studies on spheroid morphology involved observations over only 5–7 days. In the previous work of this research team, pPDLSCs were observed for 20 days to confirm their long-term stability [9]. However, the long-term stability of sPDLSC spheroids was not evaluated in this study. Therefore, further investigation of the long-term stability of sPDLSC spheroids is needed.

We found that 3D-cultured sPDLSCs exhibit a different gene expression profile, morphology, and physiology compared to identical 2D-cultured cells. Our results contribute to the development of optimized methods for the 3D culture of dental stem cells.

## 5. Conclusions

Supernumerary teeth, as expendable dental tissue, may be ethically less restrictive for clinical use than permanent teeth. Thus, they could be a good source of stem cells. Within the limitations of this study, we confirmed that the function of sPDLSCs varies depending on the culture method used. In addition, 3D-cultured sPDLSCs have greater stemness than 2D-cultured sPDLSCs. Our findings accelerate the development of regenerative medicine using MSCs derived from expendable tissue, including supernumerary teeth. Further studies that aim to enable the clinical application of 3D-cultured sPDLSCs are warranted.

**Author Contributions:** Conceptualization, Y.Y.J. and H.-S.L.; methodology, J.-H.J.; software, M.S.K.; validation, Y.Y.J., H.-S.L. and J.-H.J.; formal analysis, K.E.L.; investigation, O.H.N.; resources, S.-C.C.; data curation, Y.Y.J.; writing—original draft preparation, Y.Y.J. and H.-S.L.; writing—review and editing, M.S.K., K.E.L., O.H.N., J.-H.J. and S.-C.C.; visualization, Y.Y.J.; supervision, H.-S.L.; project administration, H.-S.L.; funding acquisition, H.-S.L. All authors have read and agreed to the published version of the manuscript.

**Funding:** This research was supported by the Bio & Medical Technology Development Program of the National Research Foundation of Korea (NRF-No.2019R1G1A1100082) and National Research Foundation of Korea (NRF) grant funded by the Korea government (MSIT) (No.2021R1G1A1013927).

**Institutional Review Board Statement:** Not applicable.

**Informed Consent Statement:** Not applicable.

**Data Availability Statement:** The data presented in this study are openly available.

**Acknowledgments:** We would like to thank Kwon Il-Geun who greatly supported this research.

**Conflicts of Interest:** The authors declare no conflict of interest.

## Summary of Abbreviations

| Abbreviations | Definition |
| --- | --- |
| 2D | Two-dimensional |
| 3D | Three-dimensional |
| ARS | Alizarin red S |
| ECM | Extracellular matrix |
| HIF | Hypoxia inducible factor |
| MSCs | Mesenchymal stem cells |
| NADP | Nicotinamide adenine dinucleotide phosphate |
| NADPH | Nicotinamide adenine dinucleotide phosphate hydrogen |
| PDL | Periodontal ligament |
| PDLSCs | Periodontal ligament stem cells |
| sPDLSCs | periodontal ligament stem cells derived from supernumerary teeth |

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
