# Peer review of "In Vitro Characterization of Periodontal Ligament Stem Cells Derived from Supernumerary Teeth in Three-Dimensional Culture Method"

_applsci, doi:10.3390/app11136040_

Round 1

Reviewer 1 Report

First of all, I would like to recognize to you the effort to perform this study. 

When writing a scientific paper, it is important not to mention the subject (for example, in lines 18-19, 105, 132, 137), you are writing "we compared the characteristics", "we observed 2D-cultured cells", "we cultured sPDLSCs..". It is not appropiate to write in this mood, it is better to write: "characteristics were compared", "2D-cultured cells were observed"...

When you write in line 35, various human tissue, I find more correct various human tissues.

I find this article well-written, but I find necessary to talk about the clinical importance of all this investigation, not only in vitro information, in order to understand it properly, because if not, it could be very technical for readers.

How many teeth have been used? where have they been obtained from??

Are supernumerary teeth frequent? Because it is maybe appropiate to know if this cells can be obtained from third molars also? I do not think supernumerary teeth are very frequent.

Besides, I think it is important to have objectives when a study is being performed. Conclusions should answer all these objectives. I would like to read it in the article.

Author Response

Reviewer 1

First of all, I would like to recognize to you the effort to perform this study. 

When writing a scientific paper, it is important not to mention the subject (for example, in lines 18-19, 105, 132, 137), you are writing "we compared the characteristics", "we observed 2D-cultured cells", "we cultured sPDLSCs..". It is not appropiate to write in this mood, it is better to write: "characteristics were compared", "2D-cultured cells were observed"...

  • Thank you. The sentences have been modified in scientific form.

When you write in line 35, various human tissue, I find more correct various human tissues.

  • Thank you. We corrected it.

I find this article well-written, but I find necessary to talk about the clinical importance of all this investigation, not only in vitro information, in order to understand it properly, because if not, it could be very technical for readers.

  • The clinical importance of this study is the use of discarded tissue (supernumerary tooth) for tissue engineering. The past research of our research team has shown the potential of 3D culture in tissue engineering. The cells used in the previous study were collected from the periodontal ligament of the permanent teeth, and it is difficult to collect them from healthy adults. The cells used in this study were obtained from the periodontal ligament of the supernumerary teeth. Supernumerary teeth are tissues that are not necessary for human body, and are usually extracted and discarded.  

How many teeth have been used? where have they been obtained from??

  • The cells used in this study were purchased. Line 86 “sPDLSCs were purchased from Cell Engineering for Origin (Seoul, South Korea).”

Are supernumerary teeth frequent? Because it is maybe appropiate to know if this cells can be obtained from third molars also? I do not think supernumerary teeth are very frequent.

  • Thank you for your comment. Both supernumerary teeth and third molars have their own pros and cons, but as a pediatric dentist, I chose the supernumerary teeth that are commonly encountered in pediatric dentistry. Depending on the studies, the prevalence of the supernumerary teeth is about 3% and most are extracted around the age of 7 or 8. On the other hand, the third molars are extracted in the late teens and may be damaged during surgery.

Besides, I think it is important to have objectives when a study is being performed. Conclusions should answer all these objectives. I would like to read it in the article.

  • Thank you for pointing out an important point. I made the object clear, and described it in conclusion.

Reviewer 2 Report

The manuscript submitted to Applied Sciences entitled “In vitro characterization of periodontal ligament stem cells derived from supernumerary teeth in three-dimensional culture method” is an original article which aim to investigate the characteristics of periodontal ligament stem cells (PDLSCs) derived from supernumerary teeth cultured using a three-dimensional method compared to a conventional two-dimensional method.

On my opinion the article is interesting, well written, with good English.

However, I highlighted some issues.

  • Title: Please use Italic for In vitro. Please do not use capital for the first letter of each word.
  • English language. Minor spell check is required.
  • Please structure the abstract to attract the reader's attention and adapt it accordingly.
  • Please improve this section. I suggest inserting the following sentence and reference regarding periodontal: << The periodontal ligament (PDL) represents a fibrous network connecting the cementum of the tooth root and the alveolar bone. It serves many functions, such as tooth support, nutrition, and protection [https://doi.org/10.1177/0963689718807680]>>.
  • Materials and Methods: Please insert ethical approval of reference institutional review board.
  • This section has been properly prepared.
  • Please discuss the effect of different culture medium on PDLSCs [https://doi.org/10.1177/0963689718807680]. Are there other similar studies that have shown similar results? Did the authors find limitations in their study by comparing it with other in the literature?
  • This section has been properly prepared.
  • Improve the quality and arrangement of figures.

Summary of abbreviations required at the end of the manuscript prior to “Reference” section.

After making the indicated changes, I am available for a second round of peer review.

Thanks for the opportunity to review this manuscript.

Author Response

Reviewer 2

The manuscript submitted to Applied Sciences entitled “In vitro characterization of periodontal ligament stem cells derived from supernumerary teeth in three-dimensional culture method” is an original article which aim to investigate the characteristics of periodontal ligament stem cells (PDLSCs) derived from supernumerary teeth cultured using a three-dimensional method compared to a conventional two-dimensional method.

On my opinion the article is interesting, well written, with good English.

However, I highlighted some issues.

  • Title: Please use Italic for In vitro. Please do not use capital for the first letter of each word.
  • Thank you. We corrected the title as you guided.

  • English language. Minor spell check is required.
  • Thank you. We checked the spell again.

  • Please structure the abstract to attract the reader's attention and adapt it accordingly.
  • Thank you. We corrected the title as you guided.

  • Please improve this section. I suggest inserting the following sentence and reference regarding periodontal: << The periodontal ligament (PDL) represents a fibrous network connecting the cementum of the tooth root and the alveolar bone. It serves many functions, such as tooth support, nutrition, and protection [https://doi.org/10.1177/0963689718807680]>>.

  • The reference was inserted.

  • Materials and Methods: Please insert ethical approval of reference institutional review board.
  • Cells used in this study were purchased and we did not need to be approved by the IRB.

  • This section has been properly prepared.
  • Please discuss the effect of different culture medium on PDLSCs [https://doi.org/10.1177/0963689718807680]. Are there other similar studies that have shown similar results? Did the authors find limitations in their study by comparing it with other in the literature?
  • In the previous study of our research team, we compare the PDLSCs from the permanent teeth cultured in 2D and 3D culture medium [ref. 9]. It had shown the potential of 3D culture in tissue engineering. Then we experimented the PDLSCs from the supernumerary teeth.

  • This section has been properly prepared.
  • Improve the quality and arrangement of figures.
  • The figures were modified.

Summary of abbreviations required at the end of the manuscript prior to “Reference” section.

  • Thank you. Summary of abbreviations were added.

After making the indicated changes, I am available for a second round of peer review.

Thanks for the opportunity to review this manuscript.

Round 2

Reviewer 2 Report

After the changes made, the article is suitable for publication.